# GRAPES: LEARNING TO SAMPLE GRAPHS FOR SCALABLE GRAPH NEURAL NETWORKS

## ABSTRACT

Graph neural networks (GNNs) learn the representation of nodes in a graph by aggregating the neighborhood information in various ways. As these networks grow in depth, their receptive field grows exponentially due to the increase in neighborhood sizes, resulting in high memory costs. Graph sampling solves memory issues in GNNs by sampling a small ratio of the nodes in the graph. This way, GNNs can scale to much larger graphs. Most sampling methods focus on fixed sampling heuristics which may not generalize to different structures or tasks. We introduce GRAPES, an *adaptive* graph sampling method that learns to identify sets of influential nodes for training a GNN classifier. GRAPES uses a GFlowNet to learn node sampling probabilities given the classification objectives. We evaluate GRAPES across several small- and large-scale graph benchmarks and demonstrate its effectiveness in accuracy and scalability. In contrast to existing sampling methods, GRAPES maintains high accuracy even with small sample sizes and, therefore, can scale to very large graphs. Our code is publicly available at https://anonymous.4open.science/r/GRAPES.

## 1 INTRODUCTION

We represent data with graph structures in many applications. Some examples are recommender systems, social networks, and the chemical and medical domains (Nettleton, 2013; Wu et al., 2022; Li et al., 2022). In these domains, graph neural networks (GNNs) are a powerful tool for representation learning on graphs (Kipf & Welling, 2016; Velickovic et al., 2017; Yun et al., 2019).

Unlike traditional machine learning on i.i.d. data, scalability is a big challenge in machine learning on graphs. One of the reasons is the connectivity of the nodes and their dependence on their neighbors, which makes dividing the data into mini-batches a non-trivial task. Secondly, in each layer of the GNN, the neighbors of the nodes in the previous layer are added to the network. Therefore, the neural network's receptive field increases with the network's depth, resulting in an exponential growth in the number of nodes it needs to process.

Graph sampling tackles the scalability problem in GNNs by considering only a sampled subset of the nodes in the graph. Currently most popular sampling methods are non-adaptive, sampling nodes independently of their features or the training task. Here, we argue for the benefits of *adaptive sampling*, where we let the sampling method adapt to the task and directly sample the nodes that lead to a highly accurate GNN. In particular, we develop a sampler that learns to sample effectively by maximizing GNN accuracy given a fixed sampling budget, i.e., the number of nodes it samples. Most current graph sampling methods instead focus on accurately approximating the behavior of the full-batch GNN (Chen et al., 2018b; Zou et al., 2019; Zeng et al., 2019; Huang et al., 2018), opting for an indirect approach to achieve high accuracy.

To this end, we develop **G**FlowNet **Grap**h **N**eighbor **S**ampling (GRAPES), an adaptive graph sampling method that uses GFlowNets (Bengio et al., 2021a). Figure 1 shows the general framework of GRAPES. The GFlowNet sampler in GRAPES gets feedback from the GNN to learn scores over the nodes to determine which should be more likely to be included. The GFlowNet framework allows us to consider the sampling process and the GNN computation together and train them in concert. This allows training a GFlowNet sampler that explicitly depends on factors like the features of the nodes, the structure of the graph, sample size, and other contextual features.

Figure 1: A high-level schematic diagram of GRAPES for a mini-batch of a two-layer classification GNN. GRAPES processes a target node (in green) by retrieving 1-hop neighbors and computing node inclusion probabilities (shown by node color shade) via a GFlowNet. Given these probabilities, GRAPES samples $k$ nodes. This process is repeated over nodes in the 2-hop neighborhood, with each step determining a state $s_i$. The sampled subgraph is passed to a GNN for target node classification. Classification loss is used to update the GNN classifier and as a reward for the GFlowNet.

We evaluate GRAPES on several node classification tasks and find that

1. GRAPES outperforms state-of-the-art sampling-based methods on most of the datasets.

2. GRAPES is competitive with a state-of-the-art non-sampling GNN scaling method, while using up to an order of magnitude less GPU memory.

3. GRAPES maintains high accuracy at low sample sizes, while performance often drops sharply for the baselines as the sample size is decreased. These results highlight GRAPES' ability to correctly identify influential nodes in the graph and allow scaling to very large graphs while keeping a high performance for the GNN.

4. GRAPES learns to distinguish between nodes and learns different preferences over nodes based on their influence on the performance of the GNN.

## 2 RELATED WORK

### 2.1 SCALABLE GRAPH LEARNING

**Fixed Policy:** In this category the sampling policy is independent of the training of the GNN and is based on a fixed heuristic that does not involve any training. Given a set of target nodes, i.e., the nodes that are to be classified, node-wise sampling methods sample a give amount of nodes for each target node. GraphSage (Hamilton et al., 2017) is a node-wise sampling method that randomly samples nodes. However node-wise sampling results in redundantly sampling one node multiple times because it can be the neighbor of several nodes (Zou et al., 2019). A more efficient approach is layer-wise sampling, for example, FastGCN (Chen et al., 2018b) and LADIES (Zou et al., 2019). They aim to minimize variance by sampling nodes in each layer with probabilities proportional to their degree. Moreover, some techniques focus on sampling subgraphs in each mini-batch, like GraphSAINT (Zeng et al., 2019) and ClusterGCN (Chiang et al., 2019). While these techniques are successful in scaling GNNs to larger graphs, they do not adapt to the GNN's performance on the task as they fix the sampling policy. In graph signal processing community, the authors in Geng et al. (2023) propose a node sampling technique based on Anis et al. (2016) that aims to sample nodes for a unique and stable graph signal reconstruction. However, this method is only applied on small graphs.

**Learnable Policy:** A few methods learn the probability of including a node based on feedback from the GNN. AS-GCN (Huang et al., 2018) is a method that learns a linear function that estimates the node probabilities in a layer-wise manner. Similarly, PASS (Yoon et al., 2021) learns a compositional policy of random and importance-based sampling, including a learnable transformation matrix to learn the similarities between the target node and the candidate neighbors. GNN-BS (Liu et al., 2020) formulates the node-wise sampling problem as a bandit problem and updates the sampling policy according to a reward function that reduces the sampling variance. SubMix (Abu-El-Haija et al., 2023) proposes a mixture distribution of sampling heuristics with learnable mixture weights. DSKReG (Wang et al., 2021) learns the relevance of items in a user-item knowledge graph by jointly optimizing the sampling strategy and the recommender model. The majority of these methods focus on variance reduction and fail to consider the classification loss, unlike GRAPES. We argue that

adaptivity to the classification loss, allows for sampling the influential nodes depending on the task, and results in better performance.

**Other Scalable Methods:** Authors of Ruiz et al. (2023) propose a method to transfer the weights of a GNN trained on a mid-sized graph to larger graphs given the graphon similarity between the graphs. Another group of papers use historical embeddings of the nodes when updating the target nodes' embeddings (Chen et al., 2018a; Fey et al., 2021; Yu et al., 2022; Shi et al., 2023). GAS (Fey et al., 2021) approximates the embeddings of the 1-hop neighbors using the historical embeddings of those nodes learned in the previous training iterations. These methods reduce the GPU memory usage by training in mini-batches and learning from the 1-hops neighbors with the historical embeddings saved in CPU memory. Unlike GRAPES, they process the whole graph.

## 2.2 GENERATIVE FLOW NETWORKS

Generative Flow Networks (GFlowNets) (Bengio et al., 2021a;b) are generative models that sample from a very large structured space. GFlowNets construct the structure in multiple generation steps. Compared to Reinforcement Learning approaches, GFlowNets learn to sample *in proportion* to a given reward function, while in Reinforcement Learning, reward functions are maximized. This feature of GFlowNets encourages sampling diverse sets of high-quality structures, instead of only considering the single best structure. GFlowNets have been used in several applications like molecule design and material science (Bengio et al., 2021a; Gao et al., 2022; Jain et al., 2022), Bayesian structure learning (Deleu et al., 2022), scientific discovery (Jain et al., 2023), and GNN explainability (Li et al., 2023). Similar to our work, the latter utilizes a GFlowNet to sample subgraphs. However, this method *explains* a trained GNN and is not used to scale GNN training to large graphs.

## 3 BACKGROUND

### 3.1 GNN TRAINING AND SAMPLING

Let $\mathcal{G}_C = (\mathcal{V}, \mathcal{E})$ be an undirected graph with a set of $N$ nodes $\mathcal{V}$ and edges $\mathcal{E}$. The adjacency matrix $A \in \{0, 1\}^{N \times N}$ indicates a connection between a pair of nodes. $\hat{A} = \tilde{D}^{-1}\tilde{A}$ is the row-normalized adjacency matrix with self-loops, where $\tilde{A} = A + I$ and where $\tilde{D}$ is the degree matrix of $\tilde{A}$. While our method is independent of the choice of GNN architecture, we limit our discussion to the GCN architecture for simplicity (Kipf & Welling, 2016). $X \in \mathbb{R}^{N \times f}$ indicates the given features on the nodes[1] and $Y \in \mathbb{Z}^{N^t}$ is the labels for target nodes $\mathcal{V}^t \subset \mathcal{V}$, the nodes with the labels. The task of node classification aims to learn the labels of these target nodes.

The output of the $l$-th layer of the GCN is $H^l = \sigma(\tilde{A}H^{l-1}W^l)$, where $W^l \in \mathbb{R}^{d^l \times d^{l+1}}$ is the weight matrix of layer $l$ and $\sigma$ is a non-linear activation function. $d^l$ is the hidden dimension of layer $l$, where $d^0$ is the input feature size and $d^L$ is the number of classes, and $L$ is the number of layers. For a node $v_i \in \mathcal{V}$, this corresponds to the update

$$h_{v_i}^l = \sigma\left(\sum_{v_j \in \mathcal{N}(v_i)} \hat{A}_{v_i, v_j} h_{v_j}^{l-1} W^l\right), \tag{1}$$

where $\mathcal{N}(v_i)$ is the set of $v_i$'s neighbors, and $\hat{A}_{v_i, v_j}$ refers to element $(v_i, v_j)$ of $\hat{A}$.

As the number of layers increases, the computation of the embedding of node $v_i$ involves neighbors from further hops. As a result, the neighborhood size grows exponentially with the number of layers. We study how to sample the graph to overcome this exponential growth. We focus on layer-wise sampling, an effective graph sampling approach (Chen et al., 2018b; Zou et al., 2019; Huang et al., 2018). First, the target nodes are divided into mini-batches of size $b$. Then, in each layer, $k$ nodes get sampled among the neighbors of the nodes in the previous layer. Therefore, the approximation of the $l$-th update of node $v_i$ is:

$$\tilde{h}_{v_i}^l = \sigma\left(\sum_{v_j \in K^l} \hat{A}_{v_i, v_j}^l \tilde{h}_{v_j}^{l-1} W^l\right), K^l \sim q(K^l | \mathcal{N}(K^{l-1})) \tag{2}$$

---

[1]If no features are available, the feature vectors are set to one-hot vectors as node identifiers.

where 1) $K^l \subseteq \mathcal{N}(K^{l-1})$ is the set of sampled nodes in layer $l$, and $\mathcal{N}(K^{l-1})$ indicates the set of neighbors of the nodes in $K^{l-1}$, 2) $K^0$ is the set of mini-batch target nodes, 3) $\hat{A}'^l(v_i, v_j)$ is the row normalized value of the sampled adjacency matrix $A'^l$ in layer $l$ [2], where $A'^l(v_i, v_j)$ is 1 if $v_j$ is sampled, i.e., $v_j \in K^l$ and 0 otherwise, and 4) $q(K^l|\mathcal{N}(K^{l-1}))$ is the probability of sampling the set of nodes $K^l$ given the nodes that were sampled in the previous layer and their neighbors. Our goal is to learn the distribution $q$ that maximizes the classification accuracy given a fixed sampling budget of $k$ nodes per layer.

## 3.2 GFlowNet and Trajectory Balance Loss

We next give a brief overview of GFlowNets (Bengio et al., 2021a;b) and the trajectory balance loss (Malkin et al., 2022a). Let $\mathcal{G}_F = (\mathcal{S}, \mathcal{A}, \mathcal{S}_0, \mathcal{S}_f, R)$ denote a GFlowNet learning problem. Here, $\mathcal{S}$ is a finite set of states that forms a directed graph with $\mathcal{A}$, a set of directed edges representing actions or transitions between states. $\mathcal{S}_0 \subset \mathcal{S}$ is the set of initial states, $\mathcal{S}_f \subset \mathcal{S}$ is the set of terminating states,[3] and $R : \mathcal{S}_f \rightarrow \mathbb{R}_+$ is the reward function defined on terminating states. At time $t$, a particular $a_t \in \mathcal{A}$ indicates the action taken to transition from state $s_t$ to $s_{t+1}$. A trajectory $\tau$ is a path through the graph from an initial state $s_0$ to a terminating state $s_n \in \mathcal{S}_f$: $\tau = (s_0 \rightarrow ... \rightarrow s_n)$. A GFlowNet is a neural network that learns to transition from an initial state $s_0$ to a terminating state where the reward $R(s_n)$ is given. The goal of the GFlowNet is to ensure that following the forward transition probabilities $P_F(s_{t+1}|s_t)$ leads to final states $s_f \in \mathcal{S}_f$ with probability in proportion to the reward $R$ (Bengio et al., 2021a). The *Trajectory Balance (TB)* loss (Malkin et al., 2022a) is developed with this goal. For a trajectory $\tau = (s_0 \rightarrow ... \rightarrow s_n)$, the TB loss is:

$$\mathcal{L}_{TB}(\tau) = \left( \log \frac{Z(s_0) \prod_{t=1}^n P_F(s_t|s_{t-1})}{R(s_n) \prod_{t=1}^n P_B(s_{t-1}|s_t)} \right)^2, \tag{3}$$

where $Z : \mathcal{S}_0 \rightarrow \mathbb{R}_+$ computes the total flow of the network from starting state $s_0$ and $P_F$ and $P_B$ are the forward and the backward transition probabilities between the states, where both can be parameterised by a neural network (Malkin et al., 2022a).

## 4 GFlowNet Graph Neighbor Sampling (GRAPES)

In this section, we introduce GFlowNet Graph Neighbor Sampling (GRAPES). GRAPES is a layer-wise and layer-dependent sampling method. In each layer $l$, it samples $k \ll |\mathcal{N}_{K^{l-1}}| = n$ neighbors of the nodes that were sampled in the previous layer. In the remainder of this section, we describe our GFlowNet design, training and reward function (Section 4.1), adjustments to the reward function (Section 4.2, and our scalable sampling procedure (Section 4.3.

## 4.1 GFlowNet Design: States, Actions, and Reward

Next, we explain our choice of $\mathcal{G}_F$, that is, the states, actions, terminating states, and reward function, and the coupling of our GFlowNet with the $GCN_C$. A state $s \in \mathcal{S}$ represent a sequence of adjacency matrices $s = (A_0, ..., A_l)$ sampled so far. An action from $s$ represents choosing $k$ nodes without replacement among the neighbors of the nodes in $A_l$. This forms the adjacency matrix $A_{l+1}$ of the next layer. Therefore, in an $L$ layer GCN, we construct a sequence of $L$ adjacency matrices to reach a terminating state.

We define the optimal sampling policy as having the lowest classification loss in expectation. Therefore, a set of $k$ nodes with a lower classification loss than another set must have a higher probability. Our goal is to design $\mathcal{G}_F$ so that it learns the forward and backward transition probabilities proportional to a given reward. We define the reward function as below:

$$R(s_L) = R(A_0, \dots, A_L) := \exp(-\alpha \cdot \mathcal{L}_{GCN_C}(A_0, \dots, A_L)), \tag{4}$$

---

[2]Unlike the full-batch GCN, when sampling, the adjacency matrix of each layer is different from the other layers because the nodes involved in each layer are different.

[3]Technically, GFlowNets have unique source and terminal (or 'sink') states $s_s$ and $s_f$. The source state has an edge to all initial states, and all terminating states have an edge to the terminal state.

where $\mathcal{L}_{GCN_C}$ is the classification loss and $\alpha$ is a *scaling parameter*, which we explain in Section 4.2. $A_0$ is the adjacency matrix of the graph given as input at the starting state, and $A_1$ to $A_L$ are the adjacency matrices sampled in each transition. We explain the sampling procedure in section 4.3.

**Forward Probability**. The GFLowNet of GRAPES is a neural network that learns the forward probabilities $P_F(s_l|s_{l+1})$, the probability of sampling an adjacency matrix $A_{l+1}$ for the layer $l+1$ given the adjacency matrices of the previous layers. Precisely, the GFlowNet learns to predict the probability $p_i$, for $i \in \{1, ..., n\}$, that the node $v_i$ in the neighbourhood of $K^l$ will be included in $\mathcal{V}^{l+1}$, the nodes sampled for layer $l+1$. We use a Bernoulli distribution to define the nodes probability inclusion in the sample. Therefore, the forward probability between layers $l$ and $l+1$ is

$$P_F(s_{l+1}|s_l) = \prod_{v_i \in \mathcal{V}^{l+1}} p_i \prod_{v_i \in \mathcal{N}(K^l) \backslash \mathcal{V}^{l+1}} (1 - p_i) \tag{5}$$

The first product computes the probabilities of the nodes that are included in the new sampled set $\mathcal{V}^{l+1}$, and the second the probabilities of the nodes that are not included.

**Backward Probability**. Trajectory balance (Equation 3) also requires defining the probability of transitioning backwards through the states. The backward probability is a distribution over all parents of a state. This distribution is not required in our setup, as the state representation $s = (A_0, ..., A_l)$ saves the trajectory taken through $\mathcal{G}_F$ to get to $s$. This means the graph for the GFlowNet learning problem $\mathcal{G}_F$ is a tree as each state $s = (A_0, ..., A_l, A_{l+1})$ has exactly 1 parent $s' = (A_0, ..., A_l)$. Since each state has a single parent, we find that $P_B(s_l|s_{l+1}) = 1$ when we retrace the trajectory. We pass the information on when a node is added to the GFlowNet by adding an identifier to the nodes' embeddings that indicates in what layer it was sampled. We also include an identifier for the mini-batch target nodes $A_0$.

Combining our setup with the trajectory balance loss (Equation 3), the GRAPES loss is

$$\mathcal{L}_{GRAPES}(\tau) = \left( \log \frac{Z \prod_{l=1}^L P_F(s_l|s_{l-1})}{R(s_L)} \right)^2 \tag{6}$$

$$= \left( \log Z + \sum_{l=1}^L \log P_F(s_l|s_{l-1}) + \alpha \mathcal{L}_{GCN_C}(A_0, ..., A_L) \right)^2 \tag{7}$$

We model the normalizer $Z(s_0)$ in Equation 3 as a real-valued trainable parameter $\log Z$ that we update using GRAPES's loss, and so assume $Z(s_0)$ is constant to remove an additional estimation step.

## 4.2 REWARD SCALING

In our experiments, we noticed that with the bigger datasets, the GFlowNet is more affected by the log-probabilities than the reward from the classification $GCN_C$. The reason that the term $\log P_F(s_{l+1}|s_l)$ is dominant in $\mathcal{L}_{GRAPES}$ is that $\log P_F(s_l|s_{l-1}) = \sum_{i=1}^n \log p(v_i)$, where $n$ is the size of neighborhood of the nodes sampled in the previous layer $l$. Given a batch size of 256, this neighborhood can be as big as $52\,000$ nodes, resulting in summing $52\,000$ log-probabilities. The majority of the probabilities $p(v_i)$ are values close to zero, therefore, the above sum would be a large negative number. Since the loss $\mathcal{L}_{GCN_C}$ is in our experiments quite close to zero, the log-probability and its variance dominate the loss. Therefore, we add the hyper-parameter $\alpha$ to the reward and tune it in our experiments.

## 4.3 SAMPLING AND OFF-POLICY TRAINING

We next describe the sampling technique of GRAPES. We have a clear constraint on the number of nodes we want to include in training: We want to sample exactly $k$ nodes without replacement. However, our forward distribution $P_F$ contains $n$ independent Bernoulli distributions, and it is unlikely that we sample exactly $k$ nodes from this distribution. Instead, we use the Gumbel-Top-k trick (Vieira; Huijben et al., 2022) which selects a set of nodes $\mathcal{V}_k^l$ by perturbing the log probabilities randomly and taking the top-$k$ among those:

$$\mathcal{V}_k^l = \text{top}(k, \log p_1 + G_1, ..., \log p_n + G_n), \quad G_i \sim \text{Gumbel}(0, 1) \tag{8}$$

The sampling process shown above is different from faithfully sampling from the GFlowNet distribution $P_F$ as we restrict the sampling space to sets of $k$ nodes. This means we train the GFlowNet *off-policy*, since we use a different sampling distribution than the forward distribution. Importantly, Malkin et al. (2022b) showed that the trajectory balance loss is stable when learning from off-policy distributions without adjusting the objective with importance weights. If we were to use a Reinforcement Learning method like REINFORCE, we would have to compute the probability of sampling $k$ nodes from $P_F$ for importance weighting, which is intractable (Ahmed et al., 2023). Therefore, the off-policy properties of GFlowNets allow us to use the Gumbel-max trick without computing importance weights or any form of normalization. We describe this issue in detail in Appendix A.

### 4.4 GRAPES Algorithm

Algorithm 1 shows one epoch of GRAPES in pseudocode. Note that in layer $l$, to preserve the self-loops of the nodes sampled in layer $l-1$, we concatenate the nodes sampled in layer $l-1$ to the sampled nodes in layer $l$, i.e., $K^l = K^{l-1} \cup \mathcal{V}_k^l$. Therefore, we build the adjacency matrix for layer $l$ $A^l \in \mathbb{R}^{|K^{l-1}| \times |K^l|}$ by adding all the edges between $K^{l-1}$ and $K^l$.

---

**Algorithm 1** One GRAPES epoch

---

**Require:** Graph $\mathcal{G}_C$, node features $X$, node labels $Y$, target nodes $\mathcal{V}^t$, batch size $B$, sample size $k$, GCN for classification $GCN_C$, and GCN for GFlowNet $GCN_F$.
1: Divide target nodes $\mathcal{V}^t$ into batches $\mathcal{V}^b$ of size B
2: **for** each batch $\mathcal{V}^b$ **do**
3:     $K^0 = \mathcal{V}^b$
4:     Build adjacency matrix $A_0$ from $K^0$
5:     **for** layer $l = 1$ to $L$ **do**
6:         $\mathcal{V}_n^l \leftarrow \mathcal{N}(K^{l-1})$                               ▷ Get all $n$ neighbors of $K^{l-1}$
7:         $p_1, ..., p_n \leftarrow GCN_F(A_0, ..., A_{l-1})$         ▷ Compute probabilities of inclusion
8:         $G_1, ..., G_n \sim \text{Gumbel}(0, 1)$                    ▷ Sample Gumbel noise
9:         $\mathcal{V}_k^l \leftarrow \text{top}(k, \log p_1 + G_1, ..., \log p_n + G_n)$     ▷ Get $k$ best nodes (Eq. 8)
10:         $K^l = K^{l-1} \cup \mathcal{V}_k^l$                                   ▷ Add new nodes
11:         Build adjacency matrix $A_l$ from $K^l$
12:     **end for**
13:     Pass all $\hat{A}_l$ to $GCN_C$ and obtain classification loss $\mathcal{L}_{GCN_C}$ given $\{(X_j, Y_j)\}_{j=1}^B$
14:     $R(A_0, ..., A_L) \leftarrow \exp(-\alpha \cdot \mathcal{L}_{GCN_C})$              ▷ Calculate reward
15:     Compute GRAPES loss $\mathcal{L}_{GRAPES}$ from Eq. 6
16:     Update parameters of $GCN_F$ by minimizing $\mathcal{L}_{GRAPES}$
17:     Update parameters of $GCN_C$ by minimizing $\mathcal{L}_{GCN_C}$
18: **end for**

---

## 5 Experiments

### 5.1 Experimental Setup

For all our experiments, we use a two-layer GCN with ReLU activation function for both classification and GFlowNet networks, $GCN_C$ and $GCN_F$. We evaluate GRAPES on the following datasets for node classification task: citation networks (Cora, Citeseer, Pubmed with the "full" split) (Sen et al., 2008; Yang et al., 2016), Reddit (Hamilton et al., 2017), Flickr (Zeng et al., 2019), Yelp (Zeng et al., 2019), and two open graph benchmarks ogbn-arxiv and ogbn-products (Hu et al., 2020). Statistics about the datasets can be found in Appendix C. We conduct our experiments on a machine with an Nvidia RTX A6000 48 GB GPU.

### 5.2 Baselines Evaluation Protocol

We compare GRAPES with the following baselines: FastGCN (Chen et al., 2018b), LADIES (Zou et al., 2019), GraphSAINT (Zeng et al., 2019), GAS (Fey et al., 2021), and AS-GCN (Huang et al., 2018). Our evaluation protocol consists of two steps: 1) training a GCN classifier via sampling and 2)

Table 1: F1-scores (%) for different sampling methods for a batch size of 256, and 256 samples. We report the mean and standard deviation over 10 runs. The best values are in **bold**, and the second best are underlined. OOM stands for "out-of-memory", which signifies that the sampling process required more GPU memory than initially allocated.

| Method | Cora | Citeseer | Pubmed | Reddit | Flickr |
|---|---|---|---|---|---|
| FastGCN | $74.93 \pm 6.62$ | $53.26 \pm 12.39$ | $53.73 \pm 23.05$ | $38.02 \pm 6.26$ | $41.06 \pm 6.50$ |
| LADIES | $69.90 \pm 18.43$ | $67.85 \pm 8.43$ | $56.36 \pm 25.29$ | $81.88 \pm 3.87$ | $43.23 \pm 0.73$ |
| GraphSAINT | $86.20 \pm 0.85$ | $77.63 \pm 0.54$ | $83.07 \pm 0.89$ | $80.50 \pm 1.00$ | $\mathbf{48.47 \pm 1.05}$ |
| AS-GCN | $\underline{86.42 \pm 0.31}$ | $\mathbf{78.80 \pm 0.21}$ | $\underline{89.76 \pm 0.53}$ | $\underline{92.44 \pm 0.56}$ | $48.21 \pm 1.70$ |
| GRAPES (ours) | $\mathbf{87.29 \pm 0.15}$ | $\underline{78.74 \pm 0.18}$ | $\mathbf{90.11 \pm 0.24}$ | $\mathbf{93.68 \pm 0.06}$ | $47.33 \pm 1.41$ |

| | Yelp | ogbn-arxiv | ogbn-products | Average F1 | Average Rank |
|---|---|---|---|---|---|
| FastGCN | $26.42 \pm 2.05$ | $29.91 \pm 3.57$ | $33.01 \pm 4.03$ | $43.79$ | $4.625*$ |
| LADIES | $16.02 \pm 2.28$ | $44.58 \pm 8.12$ | $\underline{67.72 \pm 1.95}$ | $55.94$ | $3.875*$ |
| GraphSAINT | $33.13 \pm 0.94$ | $53.71 \pm 2.79$ | $59.57 \pm 0.65$ | $65.29$ | $2.750$ |
| AS-GCN | $\underline{30.99 \pm 1.21}$ | $\mathbf{65.95 \pm 1.91}$ | OOM | $\underline{70.37}$ | $\underline{2.250}$ |
| GRAPES (ours) | $\mathbf{44.91 \pm 0.08}$ | $\underline{64.54 \pm 0.51}$ | $\mathbf{73.65 \pm 0.21}$ | $\mathbf{72.53}$ | $\mathbf{1.500}$ |

evaluating the classifier's performance on the test set using the full graph. We implement GRAPES and GraphSAINT using PyTorch Geometric (Fey & Lenssen, 2019) and refer to the source code of the original paper for the remaining baselines. The original papers evaluated the baseline methods under different configurations of GCN classifier architecture, data pre-processing, and regularization. To provide a fair comparison and study the behaviour of the sampling methods in isolation, we modify the baselines so that all the methods, only differ in the sampling technique. For more information about the adjustments in the baseline refer to Appendix B.

For selecting the learning rate, we refer to the original papers for the baselines. GRAPES requires selecting two additional hyperparameters: the learning rate for the GFlowNet and the scaling parameter $\alpha$. We tune these based on classification performance on the validation set. These hyperparameters are specific to our sampling method and do not explicitly affect the learning capacity of the GCN classifier. We refer the reader to Appendix B for details on hyperparameter settings.

## 5.3 RESULTS

**Comparison with sampling methods** We present the F1 scores of GCNs trained via sampling for GRAPES and the sampling-based baselines in Table 1. We observe that GRAPES often outperforms all these baselines. Unlike the other methods, GRAPES ranks consistently within the top 3 methods, and its average rank is the highest, suggesting that its ability to adapt the sampling policy to particular features of the data allows GRAPES to maintain consistent performance across datasets. We probe into the observed differences in rank by performing two statistical tests, using the ranks obtained for each sampling method over all datasets. First, a Friedman test reveals significant differences in ranks between all methods. We subsequently apply an all-pairs comparison and find that GRAPES ranks significantly higher than FastGCN and LADIES. AS-GCN, the next high-rank method, only significantly differs from FastGCN. The Appendix D provides more detail on our methodology for hypothesis testing.

An additional desirable property we observe is the low variance in the performance of GRAPES. After executing each experiment 10 times, we note that GRAPES has the lowest standard deviation, except for two datasets, unlike FastGCN and LADIES, which exhibit high variance. Our results demonstrate that GRAPES offers robust performance, maintaining consistency over different datasets and the randomness involved in multiple runs. FastGCN and LADIES, which use a fixed policy, fail to compete with the other methods in our experimental setup. They assign a higher probability to nodes with higher degrees. This heuristic results in neglecting informative low-degree nodes. As the results show, GraphSAINT fails to outperform the more competitive baselines.

AS-GCN is the only adaptive sampling method among the baselines and is the next best-performing method in this category, indicating the importance of adaptive sampling. However, unlike GRAPES, this method is only adaptive to the sampling variance. Furthermore, it uses a large amount of memory

Table 2: Comparison of F1-Scores (%) for GAS, and GRAPES-32 and GRAPES-256 which correspond to the different sampling sizes. We report the mean and standard deviation for 10 repeats. The best values are in **bold**.

|  | **Cora** | **Citeseer** | **Pubmed** | **Reddit** |
|---|---|---|---|---|
| GAS | $87.67 \pm 0.21$ | $\mathbf{79.37 \pm 0.35}$ | $87.94 \pm 0.31$ | $\mathbf{94.83 \pm 0.83}$ |
| GRAPES-32 | $\mathbf{88.10 \pm 0.14}$ | $79.04 \pm 0.17$ | $89.58 \pm 0.17$ | $92.43 \pm 0.13$ |
| GRAPES-256 | $87.29 \pm 0.15$ | $78.74 \pm 0.18$ | $\mathbf{90.11 \pm 0.24}$ | $93.68 \pm 0.06$ |
|  | **Flickr** | **Yelp** | **ogbn-arxiv** | **ogbn-products** |
| GAS | $\mathbf{51.32 \pm 0.30}$ | $33.79 \pm 0.61$ | $\mathbf{69.38 \pm 0.21}$ | $\mathbf{75.12 \pm 0.17}$ |
| GRAPES-32 | $45.69 \pm 1.33$ | $44.50 \pm 0.26$ | $64.03 \pm 0.57$ | $73.62 \pm 0.16$ |
| GRAPES-256 | $47.33 \pm 1.41$ | $\mathbf{44.91 \pm 0.08}$ | $64.54 \pm 0.51$ | $73.65 \pm 0.21$ |

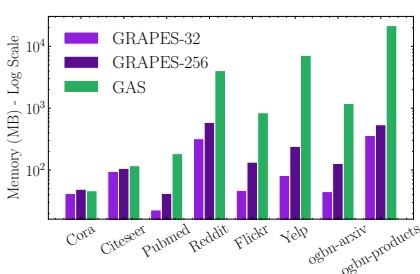

Figure 2: GPU peak memory allocation (MB) for GAS, and GRAPES-32 and GRAPES-256.

because it utilizes an attention mechanism during sampling. Therefore, AS-GCN causes out-of-memory errors when training on the largest dataset, ogbn-products, indicating AS-GCN's failure to scale up to large graphs.

**Comparison with a non-sampling scalable method** GAS (Fey et al., 2021) is a non-sampling method that uses the historical embeddings of the 1-hop neighbors of the target nodes. For GRAPES, we report the values for sampling sizes 32 and 256. We present classification results in Table 2, and we visualize memory usage in Fig. 2. While GAS has higher classification F1-scores for some datasets, GRAPES achieves comparable F1-scores and significantly outperforms GAS for Yelp. The memory usage results show that even with a large sample size, GRAPES can use up to an order of magnitude less GPU memory than GAS, especially for large datasets. In large, densely connected graphs such as Reddit, the 1-hop neighborhood can be very big. Then, the difference in memory use for GRAPES, which only sees a small set of neighbors, and GAS, which uses all the neighbors, is noticeable. While GAS occasionally achieves higher F1-scores, it consistently demands more memory, indicating a potential compromise between accuracy and computational efficiency. In contrast, both variants of GRAPES variants strike a balance, delivering comparable F1 scores with more modest memory footprints.

**GRAPES is robust to low sample sizes.** Figure 3 shows the effects of changing the sample size for all the methods except for GAS, which does not use any sampling and learns from a fixed number of nodes. To quantify the robustness to changes in the sample size, we measure deviations from the mean F1-score and find that GRAPES has the smallest deviation from the horizontal line of best fit

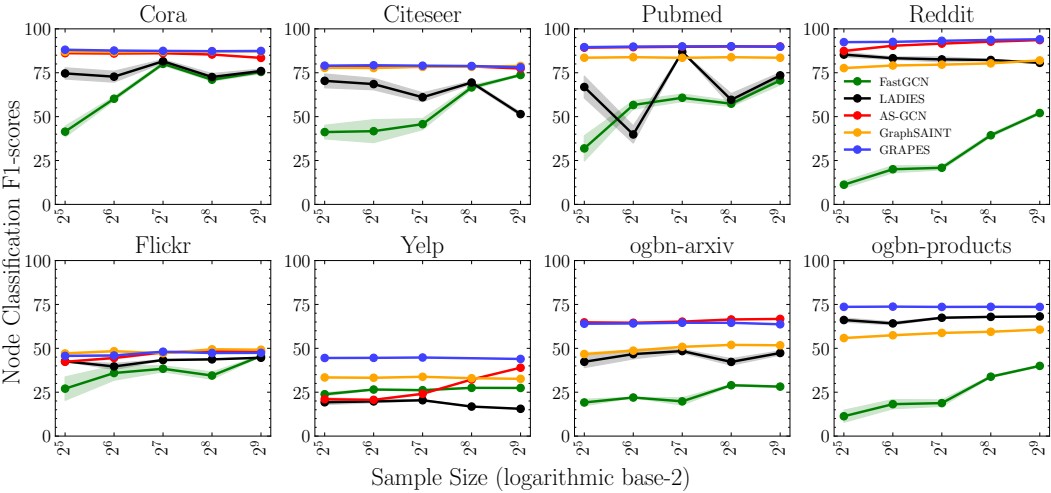

Figure 3: Comparative analysis of classification accuracy across different sampling sizes for sampling baseline and GRAPES. We repeated each experiment five times: The shaded regions show the 95% confidence intervals. AS-GCN is missing for ogbn-products due to OOM errors.

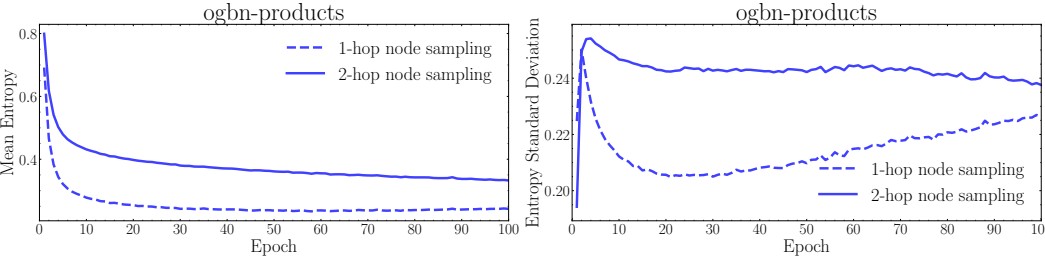

Figure 4: Entropy for the ogbn-products dataset. **Left:** Comparison of average mean entropy values over epochs. **Right:** Comparison of standard deviation entropy values over epochs.

(see Appendix E for details). This shows that the classification accuracy of GRAPES is robust to the sample size. Therefore, GRAPES achieves strong performance with fewer sampled nodes needed than the baselines, enabling training GCNs on larger graphs while using less GPU memory.

**GRAPES learns strong preferences over nodes.** The ability to selectively choose *influential* nodes is a crucial property of GRAPES. Figure 4 shows the mean and standard deviation of base 2 entropy for the node preference probabilities for the two layers of $GCN_C$ for ogbn-products. The probabilities show preference towards particular nodes with a Bernoulli distribution. A well-trained model must have a high preference (probability close to 1) for some nodes and a low preference (probability close to 0) for the rest. Therefore, we would like a low average entropy with a high standard deviation. As the figure shows, the mean entropy in both layers decreases from almost 1 and converges to a value above 0 while the standard deviation increases. This indicates that for ogbn-products, the GFlowNet initially assigned a probability near $0.5$, indicating little preference. However, after several training epochs, GRAPES starts preferring some nodes, resulting in lower mean entropy. We observe similar behavior for the large graphs (Reddit, Yelp, ogbn-arxiv, and ogbn-products). However, we observe that the GFlowNet exhibits no preferences among the nodes for the other datasets. For more details about the rest of the datasets, see Appendix F.

## 6 DISCUSSION AND CONCLUSION

We propose GRAPES, a GFlowNet-based graph sampling method that facilitates the scalability of training GNNs on very large graphs. GRAPES learns the importance of the nodes by adapting to the classification loss. We show how a GFlowNet can help build a subgraph of influential nodes by learning the node preferences adaptive to the node features, GNN architecture, classification task, and graph topology. Our experiments demonstrate that GRAPES effectively selects nodes from large-scale graphs and achieves consistent performance over state-of-the-art sampling methods. Moreover, we show GRAPES compared to the other sampling methods, can maintain a high level of classification accuracy even with lower sampling rates, indicating GRAPES' ability to scale to larger graphs by sampling a small but influential set of nodes. GRAPES achieves comparable performance to GAS, while using up to an order of magnitude less memory.

**Lack of Uniform Evaluation Protocol.** Existing methods in the literature report performance on graph sampling under settings with different GCN architectures, regularization techniques, feature normalization strategies, and data splits, among other differences. These differences made it challenging to determine the benefits of each sampling method. This motivated us to implement a unified protocol across all methods where we keep the architecture fixed. We encourage future work to consider a similar methodology for a fair evaluation, or an experimental review study, as is common in other areas of machine learning research on graphs (Shchur et al., 2018; Ruffinelli et al., 2019).

**Known Limitations.** In our experiments, we focused on the problem of node classification. However, GRAPES is not tied to a particular downstream task. In general, GFlowNets assume access to a tractable reward function (Bengio et al., 2021b). Therefore, GRAPES will be applicable for other graph-related tasks with tractable rewards, like link prediction and unsupervised representation learning. We leave these directions for future work. Furthermore, we assumed that the normalizing function $Z(s_0)$ in Equation 3 is constant across different mini-batches to reduce estimation overhead, and we leave the study on the effect of estimating $Z(s_0)$ on the sampling for future work.

**Reproducibility Statement.** Our code is publicly available at `https://anonymous.4open.science/r/GRAPES`. The full set of hyperparameters used for GRAPES are available in the repository. As described previously, we adapted the baselines (LADIES, GAS, AS-GCN, FastGCN, GraphSAINT) for a fair comparison; the link to the code and the corresponding hyperparameters for the modified baseline models can be found in our repository. The link to the code and data for the data analyses we carried out are also available in our repository.

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
