# A  OFF-POLICY SAMPLING SETUP

In this Appendix, we discuss the technical and mathematical challenges around our setup that resulted in our off-policy learning setup. In each layer $l$ of the GFlowNet, we aim to sample exactly $k$ out of $n$ nodes. An initially natural setup would be to use the distribution over $k$-subsets of $\mathcal{N}(K^l)$ (Ahmed et al., 2023). Continuing the notation in Equation 5:

$$p'(s_{l+1}|s_l) = \frac{I[|\mathcal{V}^{l+1}| = k]P_F(s_{l+1}|s_l)}{\sum_{s'_{l+1}} I[|\mathcal{V}^{l+1'}| = k]P_F(s'_{l+1}|s_l)} \tag{9}$$

where $P_F(s_{l+1}|s_l)$ is as defined in Equation 5. $p'$ assigns 0 probability to states $s_{l+1}$ that do not add exactly $k$ new nodes (that is, when $|\mathcal{V}^{l+k}| \neq k$). However, this requires renormalizing the distribution, which is the function of the denominator term on the right hand side. Note that this sum is over an exponential number of elements, namely $2^n$, and naive computation is clearly intractable. SIMPLE (Ahmed et al., 2023) provides an optimized dynamic programming algorithm for computing this normalization constant. However, it scales polynomially in $n$ and $k$, and computing the normalizer is intractable already in mid-sized graphs like Reddit.

Therefore, we decided to circumvent having to compute $p'(s_{l+1}|s_l)$ by taking the factorized distribution $P_F$ as the forward distribution and sampling using the Gumbel-Top-k trick (Equation 8) to ensure we always add exactly $k$ nodes. However, we are now in an off-policy setting: The samples using Equation 8 are distributed by $p'$, not by $P_F$, and so we sample from a different distribution than the one we use to compute the loss. Previous work (Malkin et al., 2022b) showed that the Trajectory Balance loss is amenable to off-policy training without importance sampling and weighting without introducing high variance. This is important since importance weighting would require us to weight by $P_F(s_{l+1}|s_l)/p'(s_{l+1}|s_l)$, reintroducing the need to compute $p'$.

The off-policy benefits of Trajectory Balance provide a strong argument over more common Reinforcement Learning setups. Off-policy training in Reinforcement Learning usually requires importance weighting to be stable, which is not tractable in our setting.

# B  EXPERIMENTAL DETAILS

Our experiments were carried out in a single-node cluster setup. We conducted our experiments on a machine with an Nvidia RTX A6000 GPU (48GB GPU memory) and each machine had 48 CPUs.

## B.1  HYPERPARAMETER TUNING

We tune the hyperparameters of GRAPES using a random search strategy with the goal of maximizing the accuracy of the validation dataset. We used Weights and Biases for hyperparameter tuning [4]. The best-performing hyperparameters for every dataset can be found in our repository `https://anonymous.4open.science/r/GRAPES`. The following are the hyperparameters that we tuned: the learning rate of the GFlowNet, the learning rate of the classification GCN, and the scaling parameter $\alpha$. We used the log uniform distribution to sample the aforementioned hyperparameters with the values from the following ranges respectively, $[1e-6, 1e-2]$, $[1e-6, 1e-2]$, and $[1e2, 1e6]$. We kept the other hyperparameters such as the batch size and hidden dimension of the GCN. We used the Adam optimizer (Kingma & Ba, 2014) for $GCN_C$ and $GCN_F$. The number of epochs was selected between 50, 100, and 150 depending on the performance on the validation set.

## B.2  BASELINES

For a fair comparison, we adjusted the implementations of the baselines so that the only difference is the sampling methods and the rest of the training conditions are kept the same. In the following, we explain the details of the modifications to each of the baselines.

For LADIES we used the official implementation, which also contains an implementation of FastGCN. We changed the nonlinear activation function from ELU to ReLU, and we removed any linear layers after the two layers of the GCN, set dropout to zero, and disabled early stopping. We also noticed

---

[4]`https://wandb.ai`

Table 3: Statistics of the datasets used in our experiments. The label rate indicates the percentage of nodes used for training.

| Dataset | Task | Nodes | Edges | Features | Classes | Label Rate (%) |
|---|---|---|---|---|---|---|
| Cora | multi-class | 2,708 | 5,278 | 1,433 | 7 | 44.61 |
| CiteSeer | multi-class | 3,327 | 4,552 | 3,703 | 6 | 54.91 |
| PubMed | multi-class | 19,717 | 44,324 | 500 | 3 | 92.39 |
| Reddit | multi-class | 232,965 | 11,606,919 | 602 | 41 | 65.86 |
| Flickr | multi-class | 89,250 | 449,878 | 500 | 7 | 50.00 |
| Yelp | multi-label | 716,847 | 6,977,409 | 300 | 100 | 75.00 |
| ogbn-arxiv | multi-class | 169,343 | 1,157,799 | 128 | 40 | 53.70 |
| ogbn-products | multi-class | 2,449,029 | 61,859,076 | 100 | 47 | 8.03 |

that the original LADIES implementation divided the target nodes into mini-batches, not from the entire graphs as we do, but into random fragments. This means that LADIES and FastGCN do not see all of the target nodes in the training data. We kept this setting unchanged, because otherwise it significantly slowed down the training of these two methods.

For GraphSAINT we noticed that the GNN consists of two layers of higher order aggregator, which is a combination of GraphSage-mean (Hamilton et al., 2017) and MixHop (Abu-El-Haija et al., 2019), and a linear classification layer in the end. Moreover, the original implementation of GraphSAINT is only applicable to inductive learning on the graphs, where the train graph only contains the train nodes and is entirely different from the validation and test graphs where only the nodes from the validation set and test are available respectively. We argue that in transductive learning unlike inductive learning, the motivation to scale to larger graphs is higher since the validation and test nodes are also available during training, and therefore, the processed graph is larger. Finally we used Pytorch Geometric's function for GraphSAINT node sampling to keep all the configurations the same as GRAPES. Therefore, we use the same GCN and data loader (transductive) as ours and use GraphSAINT to sample a subgraph of nodes for training. We use the node sampler setting since it is the only setting that allows specifying different sampling budgets, and therefore, can be compared to layer-wise methods with the same sampling budget. We also removed the early stopping and use the same number of epochs as GRAPES. Please refer to our repository for more details about the implementation of GraphSAINT.

For GAS, we used the original implementation. However, we changed the configuration of the GCN to have a two-layer GCN with 256 hidden units. We turned off dropout, batch normalisation and residual connections in the GCN. We also removed early stopping for the training.

For AS-GCN we removed the attention mechanism used in the GCN classifier. Their method also uses attention in the sampler, which is separate from the classifier, so we keep it.

## C  DATASET STATISTICS

We present the statistics of the datasets used in our experiments in Table 3. The splits that we used for Cora, Citeseer, and Pubmed correspond to the "full" splits, in which the label rate is higher than in the "public" splits.

## D  METHODOLOGY FOR HYPOTHESIS TESTING

In section 5.2 we present the classification performance of a GCN classifier, after training it with the sampling baselines and GRAPES. We then perform statistical tests aimed at answering two questions.

**Are the differences between ranks statistically significant?**   Our null hypothesis is that the ranks of all methods have the same distribution. We present the ranks in Table 4, together with the resulting average rank. We apply a Friedman test (Friedman, 1937) to compare the ranks of each method, obtaining a $p$-value of 0.0005. We thus reject the null hypothesis and conclude that the observed difference in rank across methods is statistically significant.

**Is the difference in ranks between specific pairs of methods statistically significant?** In this case, we are interested in multiple null hypotheses involving pairs of methods. For example, *do the ranks obtained from GRAPES and FastGCN come from the same distribution?* As we are interested in similar hypotheses involving different methods, this requires comparing multiple pairs and adjusting the $p$-value to account for multiple comparisons (Demšar, 2006). We carry out a post-hoc Nemenyi's test using the `scikit-posthocs` library (Terpilowski, 2019) and obtain the $p$-values shown in Table 5. The results show that the difference in rank between GRAPES is significant with respect to FastGCN and LADIES with $p < 0.05$.

## E   PEARSON'S CHI-SQUARED ($\chi^2$) TEST AS A ROBUSTNESS MEASURE

Figure 3 shows the effect of changing sample size on the node classification performance. In an ideal case, it would be desirable to have a constant accuracy as the number of the sample size changes. To quantify the robustness of the different sampling methods, we compute the goodness-of-fit of observed accuracy values to a horizontal line of best fit (*i.e.* a constant accuracy value). To this end, we use Pearson's chi-squared test (Pearson, 1900). Let's assume we have a set of observed accuracies, denoted as $O_i$, and a horizontal line of best fit which represents the expected accuracy, $E_i$. The chi-square statistic, $\chi^2$, is computed using the formula:

$$\chi^2 = \sum_{i=1}^{n} \frac{(O_i - E_i)^2}{E_i},\tag{10}$$

where $n$ is the number of observed values. A high $\chi^2$ value would indicate that the observed accuracies significantly deviate from the expected accuracy, while a low value would suggest that they are close. The expected accuracy $E_i$ is the average accuracy across all observations for a particular combination of method and dataset. We did not include GAS (Fey et al., 2021) as the approach does not sample any nodes.

Table 6 shows the $\chi^2$ values for the different sampling methods we study. GRAPES has the lowest average $\chi^2$ across all the different datasets, which implies that is very robust to changes in the sampling sizes compared to the baseline sampling methods.

**Rank Significance Test.** Our null hypothesis is that the ranks of all methods have the same distribution. We also performed a Friedman test to compare the ranks of each method, obtaining $p$-value of 0.000627. Therefore, we reject null hypothesis, and conclude that the ranks presented in Table 6 are statistically significant.

## F   ENTROPY AS NODE PREFERENCE MEASURE

Figures 5 and 6 show the mean and standard deviation of entropy in base two of all the datasets. We calculate the mean entropy for as the following:

$$E = \frac{1}{n} \sum_{i=1}^{n} p_i \cdot \log_2(p_i) + (1 - p_i) \cdot \log_2(1 - p_i)\tag{11}$$

Table 4: Ranks according to the F1 scores obtained by each sampling method on the node classification task. Asterisks (*) indicate that the difference in average rank to GRAPES is statistically significant at $p < 0.05$.

| Method | Cora | Citeseer | Pubmed | Reddit | Flickr | Yelp | ogbn-arxiv | ogbn-products | Average |
|--------|------|----------|--------|--------|--------|------|------------|---------------|---------|
| FastGCN | 4 | 5 | 5 | 5 | 5 | 4 | 5 | 4 | 4.625* |
| LADIES | 5 | 4 | 4 | 3 | 4 | 5 | 4 | 2 | 3.875* |
| GraphSAINT | 3 | 3 | 3 | 4 | 1 | 2 | 3 | 3 | 2.750 |
| AS-GCN | 2 | 1 | 2 | 2 | 2 | 3 | 1 | 5 | 2.250 |
| GRAPES   (ours) | 1 | 2 | 1 | 1 | 3 | 1 | 2 | 1 | **1.500** |

Table 5: $p$ values obtained with the Nemenyi post-hoc test for comparing the ranks for all pairs of methods.

| Method | FastGCN | LADIES | GraphSAINT | AS-GCN | GRAPES |
|---|---|---|---|---|---|
| FastGCN | 1.0000 | 0.8669 | 0.1233 | 0.0223 | 0.0010 |
| LADIES | 0.8669 | 1.0000 | 0.5977 | 0.2396 | 0.0223 |
| GraphSAINT | 0.1233 | 0.5977 | 1.0000 | 0.9000 | 0.5080 |
| AS-GCN | 0.0223 | 0.2396 | 0.9000 | 1.0000 | 0.8669 |
| GRAPES (ours) | 0.0010 | 0.0223 | 0.5080 | 0.8669 | 1.0000 |

Table 6: Chi-square values. Each experiment was repeated five times. The $\chi^2$ were computed with the mean accuracies. The best values are in **bold**, and the second best are underlined. OOM indicates an out-of-memory error. Asterisks (*) indicate that the difference in average rank to GRAPES is statistically significant at $p < 0.05$.

| Method | Cora | Citeseer | Pubmed | Reddit | Flickr |
|---|---|---|---|---|---|
| FastGCN | 14.4540 | 17.3168 | 14.7477 | 38.3733 | 4.9597 |
| LADIES | 0.7089 | 4.0147 | 18.6169 | 0.1393 | 0.3329 |
| GraphSAINT | **0.0016** | **0.0130** | **0.0013** | 0.1304 | 0.1004 |
| AS-GCN | 0.0602 | 0.0137 | 0.0033 | 0.2550 | 0.6420 |
| GRAPES | 0.0047 | 0.0616 | 0.0018 | **0.0209** | **0.0911** |

| | Yelp | ogbn-arxiv | ogbn-products | Average $\chi^2$ | Average Rank |
|---|---|---|---|---|---|
| FastGCN | 0.3335 | 3.6756 | 23.4954 | 14.6695 | 4.6250* |
| LADIES | 0.9420 | 0.7474 | 0.1607 | 3.2079 | 3.7500* |
| GraphSAINT | 0.0232 | 0.3967 | 0.2358 | 0.1128 | 2.0000 |
| AS-GCN | 9.2825 | 0.0599 | OOM | 1.4738 | 3.0000 |
| GRAPES | **0.0084** | **0.0085** | **0.0002** | **0.0246** | 1.6250 |

where $n$ is the number of neighbors of the nodes sampled in the previous layer and $p_i$ is the probability of inclusion for each node, which is the output of the GFlowNet. As the figures show, for the small datasets (Cora, Citeseer, Pubmed, Flickr) the mean entropy is: 1) very close to 1, indicating that GRAPES prefers every nodes with the probability close to 0.5, or 2) close to 0 but also with a low standard deviation, meaning that it equally prefers the majority of the nodes with the probability 1 or 0. On the contrary, for the large datasets (Reddit, Yelp, ogbn-arxiv, ogbn-products) by the end of training, the average entropy is lower than 1, with a standard deviation around 0.3 indicating that GRAPES learns different preferences over different nodes, some with a probability close to 1, and some close to 0.

## G   GPU MEMORY USAGE COMPARISON BETWEEN GRAPES AND GAS

We compared different variants of GRAPES, with different sample sizes (32, 256), with GAS (Fey et al., 2021), which is a non-sampling method. Figure 2 shows the GPU memory allocation (MB), in a logarithmic scale, and the F1-scores for GRAPES-32, GRAPES-256 and GAS. The three graph methods exhibit distinct performance characteristics across various datasets. We used the `max_memory_allocated` function in PyTorch to measure the GPU memory allocation.[5] Since this function measures the maximum memory allocation since the beginning of the program, where the memory measurement is done does not matter.

---

[5]`https://pytorch.org/docs/stable/generated/torch.cuda.max_memory_allocated.html`

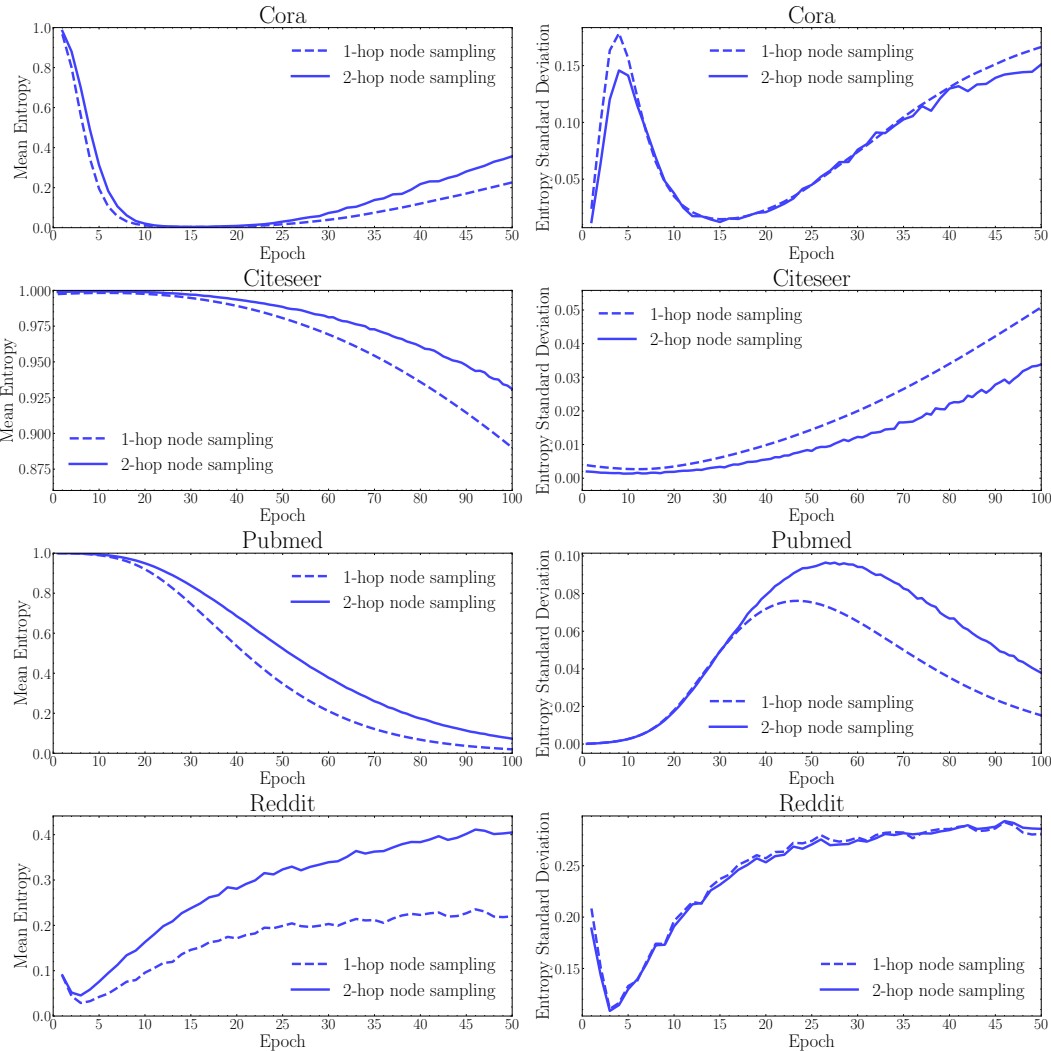

Figure 5: Combined entropy plots for Citeseer, Cora, Flickr and ogbn-arxiv, showcasing the mean entropy and entropy standard deviation across epochs. The plots compare 1-hop node sampling against 2-hop node sampling.

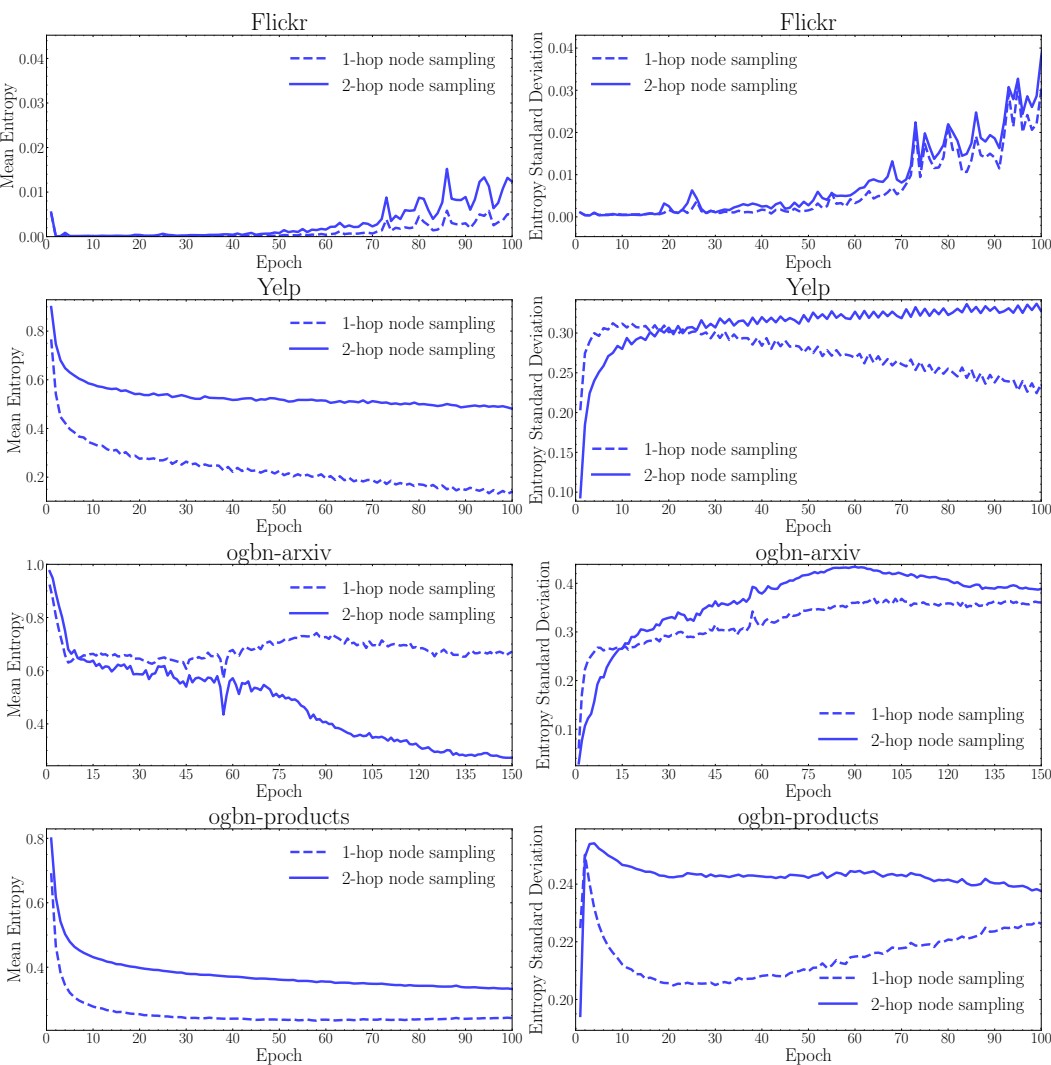

Figure 6: Combined entropy plots for ogbn-products, Pubmed, Reddit and Yelp, showcasing the mean entropy and entropy standard deviation across epochs. The plots compare 1-hop node sampling against 2-hop node sampling.