# OpenReview forum: "GRAPES: Learning to Sample Graphs for Scalable Graph Neural Networks"
_ICLR.cc/2024/Conference — Submitted to ICLR 2024_

### Official Review · Reviewer_J9DQ · 2023-11-01

**Soundness:** 3 good
**Presentation:** 3 good
**Contribution:** 2 fair
**Rating:** 6
**Confidence:** 4

**Summary:**

Paper is interested in sampling (smaller) subgraphs from (large) input graph when training Graph Neural Networks (GNNs), as a way of scaling of GNN training onto larger graphs. Unlike most earlier methods, where their sampling logic is fixed and non-trainable, the proposed method has a subgraph sampling function that is trainable. It samples subgraphs level-by-level, starting from an input batch. They sample next nodes from probability distribution over all nodes. It is conditioned on the nodes visited prior. They parameterize the distribution using GFLowNet [Bengio et al, 2021].

**Strengths:**

## Strengths
**Problem space**: sampling subgraphs from large graphs when training GNNs, which can scale GNNs to very large graphs, is important, for both static graphs and dynamic graphs.

**Novel Model definition**: they present "sampling subgraphs" as leaf-nodes of Finite State Machines (FSMs), and their FSMs look like trees (strong model assumptions*). This model definition is novel.

**Clarity of writing**. The paper is concise and up-to-the-point. Algorithm 1 is ties the pieces well.


## Summary

The novelty of construction appeals me to recommend this paper for acceptance. However, it has many small weaknesses. To do a better justice to the goodness of your work, if you have time to address all (or most) of my concerns and questions, within the main paper, I should be able to change my review.

**Weaknesses:**

Here, I point out major points that need revision. In addition, in the next **Questions** section, I ask for clarifications on more minor (but still important) issues.


## The specialization of GFLowNet onto trees be explicit stated

Above Eq. 6, it says that $P_B(s_{l} | s_{l+1}) = 1$ -- this implies that the only way to get to $s_{l+1}$ is through $s_{l}$. This strong modeling assumption stems from "s" being the entire path of generated adjacencies $\{A_0, \dots, A_l \}$. This produces a special case of "finite state machines" that specifically have states looking like trees.

## Larger graphs?

Only one graph more than 1 million nodes whereas one central theme of the paper is about scale.


## Application appeal

I would wish that the paper considers applying their method to a problem space beyond graph sampling (or otherwise, show compelling use-cases for graph sampling). Specifically, could this method be used to *explain graphs*? E.g., in the integrated-gradient sense: the presence of which nodes or edges would cause a certain prediction.

## Missing section on inference

While the paper includes the information about training, it should also include information on how to do inference. Given a node at a (large) input test graph, are samples taken or the full graph around $n$?

## Missing References on learnable sampling

E.g.,

* DSKReG; CIKM'2021
* "Performance-adaptive sampling strategy towards fast and accurate GNNs.", KDD'2021
* Submix; UAI'2023

**Questions:**

The following items are not clear. Please clarify them in the paper

Q1:
How is the $GNN_F$ parameterized? Does it train a scalar for every node (i.e., lookup 1D embedding table) or is it a function of features? In my understanding, GNN_F models $P_F$ (correct me if I am wrong) i.e. should have support on the nodes

Q2:
Is the reward measured only on end states? (on the "sum" of sampled list of adjacency matrices) or on every intermediate state (e.g., sum of adjacencies at that point).

Q3: **Runtime experiments** Would you report runtimes? E.g., on the largest dataset ogbn-products?

Q4: **Repeat edges**. Can an edge be sampled twice? Does this have any impact on the GCN model?

Q5: Is adjacency matrix $A^0$ same as $A_0$ (Algorithm 1)

Q6: $Z(s_0)$ in text following Eq. 3 -- It is not clear whether scalar $log Z$ is modeled or if it is a constant and removed.

---

> ### Author Response · Authors · 2023-11-17
> **Response to the Comments from Reviewer J9DQ**
>
> Thank you for your feedback. Here are our answers to your comments and questions.
>
> 1. The tree assumption: Indeed, the graph over which the GFlowNet acts, which can indeed be seen as a finite automata, is a tree. We have highlighted and explained this fact further just above Equation 6.
>
>    The reason we made this assumption is technical. If we do not add to the states the order when the nodes are added, we must assign some probability to all the previous states that we could come from. However, counting how many such states there are is surprisingly complex, as nodes could be added because they are 1-hop or 2-hops away from the initial target nodes. Ultimately, we do not believe this assumption will impact performance in any way.
> 2. Larger datasets: We agree with the reviewer. However, we followed the recent related work such as GAS, LMC, and LABOR [1] (NeurIPS 2023) and used the datasets they reported.  However, other datasets in our experiments were also relatively large, like Yelp, with more than 700K nodes and almost 7 million edges, and Reddit, with more than 200K nodes and 11 million edges. Note that the full-batch GCN leads to out-of-memory errors with 16GB GPU memory for Reddit, Yelp, and products.
> 3. Application appeal: This paper focuses on improving the scalability of GNNs with sampling. Other applications, such as the explainability of GNNs with GFlowNets, are orthogonal topics to sampling and require further research. These would be interesting for future work. However, adding more applications to our paper beyond sampling would require going less in-depth with the method. It may change the paper into a survey paper on the applications of GFlowNets. For other applications of GFlowNets, as mentioned in the related work, there are several studies, such as graph explainability [2], molecule design, and material science [3, 4, 5].
> 4. The inference specification: Section 5.2 mentions that we use the full graph for the evaluation, which is the same protocol all our baselines have used.
> 5. New references: Thank you for the pointers. We now included them in the related work, although the second reference, "Performance-adaptive sampling strategy towards fast and accurate GNNs.", KDD'2021, was already in the paper.
> 6. Architecture of GNN_F: The architecture of GNN_F is composed of a two-layer GCN with support on the input nodes and its neighbors at different levels, and its outputs are the sampling probabilities for each node. We do not train a scalar for every node as an input to this GCN but instead use the node features present in the datasets we experiment with.
> 7. Reward measured on the end state: Yes, in GFlowNet, the reward is given in the end state, not the middle states.
> 8. Repeated edges: In each layer, an edge cannot be sampled twice. However, one edge can be sampled in both layers. This should not have a negative effect on the GCN because in the full-batch GCN, after two layers, the target nodes are updated twice by the 1-hop neighbors and once updated by the 2-hop neighbors. Currently, all the layer-wise sampling methods in our baselines (FastGCN, LADIES, AS-GCN) follow the same regime. However, one can design a layer-wise sampling method that samples each edge only once across different layers so that in each layer, only new edges are sampled. Studying that effect is an interesting question that can be pursued in future work.
> 9. Typo for A_0: Thank you for pointing this out. We fixed this in the paper.
> 10. Value of Z: as mentioned in the lines under E.q. 7, Z is a learnable scalar, updated in each backpropagation step. However, the value of Z is kept the same for the different batches. In the paper, we refer to this point as one of the known limitations.
>
> ### References
> [1] Balın, M.F. and Çatalyürek, Ü.V., 2022. yer-neigh (BOR) Sampling: Defusing Neighborhood Explosion in GNNs. arXiv preprint arXiv:2210.13339.
>
> [2] Wenqian Li, Yinchuan Li, Zhigang Li, Jianye Hao, and Yan Pang. Dag matters! Gflownets enhanced explainer for graph neural networks. arXiv preprint arXiv:2303.02448, 2023.
>
> [3] Emmanuel Bengio, Moksh Jain, Maksym Korablyov, Doina Precup, and Yoshua Bengio.
> Flow network based generative models for non-iterative diverse candidate generation. Advances in Neural Information Processing Systems, 34:27381–27394, 2021a.
>
> [4] Wenhao Gao, Tianfan Fu, Jimeng Sun, and Connor Coley. Sample efficiency matters: a benchmark for practical molecular optimization. Advances in Neural Information Processing Systems, 35: 21342–21357, 2022.
>
> [5] Moksh Jain, Emmanuel Bengio, Alex Hernandez-Garcia, Jarrid Rector-Brooks, Bonaventure FP Dossou, Chanakya Ajit Ekbote, Jie Fu, Tianyu Zhang, Michael Kilgour, Dinghuai Zhang, et al. Biological sequence design with gflownets. In International Conference on Machine Learning, pp. 9786–9801. PMLR, 2022.

---

### Official Review · Reviewer_iUex · 2023-11-01

**Soundness:** 2 fair
**Presentation:** 4 excellent
**Contribution:** 3 good
**Rating:** 3
**Confidence:** 5

**Summary:**

The paper proposes an adaptive sampling algorithm to learn an influential subgraph in each layer of a GNN classifier. Instead of fixed heuristics, the proposed method learns the preferences between nodes via the influence they have on the overall performance of the classifier. Designed based on the GFlowNet Architecture, it identifies a sequence of states that represent a sequence of samples in each layer of the GNN. They have shown an improvement in terms of F1 score and GPU memory utilization compared to other non-adaptive algorithms like FastGCN and LADIES, and even the adaptive AS-GCN method.

**Strengths:**

1. The method seems to have an improvement on F1 scores of GRAPES on most of the datasets compared to the other algorithms in the presented experimental setup. It has also proved to consume much less memory compared to GAS which has a different non-sampling strategy to reduce the scalability problem in large graphs. Although, it outperforms GRAPES in some datasets.

1. Different types of results such as the F1 scores, GPU memory allocation, robustness and entropy are provided to demonstrate the effectiveness of the proposed algorithm.

1. The paper, in general, is well-written and sufficient for the reader to understand the concepts involved.

**Weaknesses:**

1. **Experimental Setup**: In the presented setup, the proposed method outperforms the baselines. However, a few things about the setup are not clear:
    1. It is not clear if the baselines were tuned on a validation set. Why was the batch size fixed to 256 for the main results table?
    1. A related concern is the appearance of low F1-score compared to what is reported in other paper. Granted that this is in the transductive setting, I am not sure, if that should cause such decrease in performance. For instance, according to the GraphSAINT paper, it achieves 96.6% on Reddit in the inductive setting. In this paper, the result is much lower (80.50).
    1. Comparison against GraphSAINT: Fixing 256 size and 256 samples does not seem fair for GraphSAINT. It uses a sample per minibatch instead of such a low number of samples. Also, GraphSAINT paper shows sample size of few thousands, while this paper uses sample size only up to $2^9$.  Also, the node sampler seems to have the lowest performance compared to other samplers of GraphSAINT, so other samplers should have been considered (edge, RW, Multidimensional RW).
    1. Architectures beyond GCN should be considered. If the sampling approach improves over baselines for multiple architectures such as GAT, GIN, SAGE, then it would create a strong case for the proposed sampling approach. As of now, the central claim does not seem justified, "GRAPES outperforms state-of-the-art sampling-based methods." What has been shown is that GRAPES outperforms other methods on a specific GCN architecture and under small sample sizes and number of samples.
    1. Most of the baselines presented are relatively old.

1. **Discussion on runtime** - The downside of being adaptive is that there is extra computation involved per batch. However, no discussion of training time has been presented. This would have helped with understanding execution time - F1 tradeoff.

**Questions:**

1. Were the baselines tuned on a validation set?
1. Why are the baseline performance lower compared to what is seen in other papers? I would expect the transductive setting to improve the results compared to the inductive setting.
1. How is the performance vs GraphSAINT with a higher number of samples and larger sample size?
1. Have other architectures been considered (other than GCN)?
1. Can you present a comparison of training times of the proposed approach vs the baselines?

---

> ### Author Response · Authors · 2023-11-17
> **Response to the Comments from Reviewer iUex**
>
> Thank you for your feedback. Here are our answers to your comments and questions.
>
> 1. Baselines tuned on the validation set: We used the hyperparameters reported by the baselines. For a fair comparison, we only changed the hidden dimensions so that all the methods have the same GCN hidden dimension, i.e., the same computation capacity.
> 2. Why the batch size of 256: We used this batch size because most of the baselines used this number, which required the least amount of change in the baselines and gave more trust in the hyperparameters of the baselines.
> 3. The baselines’ performance was lower than reported: Each of the baselines used different GNN architectures, sampling rates, batch sizes, etc., which makes it difficult to judge if the reason for the improvement in the accuracy is the sampling method or these differences. Therefore, for a fair comparison, we kept all the other factors equal for all the methods. For GraphSAINT, significant changes were made in the GNN architecture (from two higher-order aggregators to two GCN layers) and the number of nodes per batch (from thousands to 256+256+256 nodes). Both of these factors, especially the number of nodes, have a high effect on the final accuracy of the GNN with any sampling method. Therefore, the only way to say which method performs the best is to keep these factors the same.
> 4. Why we chose _node-sampler_ in GraphSAINT: One of the challenges we faced was determining how to compare different sampling methods, i.e., node-wise, layer-wise, and subgraph-wise because keeping the number of sampled nodes consistent is difficult across these methods. Our question is, given an equal budget of sampled nodes, which method leads to the best performance. The _node-sampler_ in graphSAINT is the only sampler in this method that allows us to specify how many nodes we want to sample. The _RW-sampler_ starts with $n$ batch nodes and does $k$ walks. Ultimately, it will sample a maximum of $n*(k+1)$ nodes. In Figure 3, we wanted to evaluate different methods when sampling fewer nodes per layer than the batch size (here $n=256$). However, with the _RW-sampler_, the minimum number of nodes sampled in each neighborhood hop is n. We have a paragraph in Appendix 2 (A2) explaining this in the paper.
> 5. Using different GNN architectures: We have tested GRAPES on several different datasets with different structures and connectivities, from single-label to multi-label nodes. GRAPES has shown superior performance to the baselines in these settings. While experiments with a different GNN architecture are interesting, including them would require re-executing all our main experiments. We consider that the fact that GRAPES improves upon the baselines under different datasets and tasks provides strong evidence of performance generalization. Nonetheless, we will include experiments with a different GNN architecture in the camera-ready. However, testing on different architectures is impossible in a week because we need to tune the parameters for GRAPES, as well as run the baselines. We will include these results in the camera-ready version. About the small sample size point, we argue that having a small sample size is more challenging than a large sample size because less data is available to learn from. Increasing the batch size will also increase memory usage, which is the main problem that graph sampling tries to overcome. Moreover, the smaller batch size allows a larger embedding size or input features. This can be especially useful in graphs with multimodal nodes that need large feature encoders.
> 6. Runtime report: A full training and testing experiment of GRAPES on our largest dataset, ogbn-products, with 100 epochs takes around 100 minutes. While other methods, such as GAS, have optimized implementations with low-level code that lead to lower runtimes, we found the runtime of GRAPES reasonably low to allow for extensive experimentation. It’s important to note that our main focus was GPU memory optimization. We achieved this goal while being able to run GRAPES on large graphs within a feasible runtime. In the future, we plan to explore optimizations such as using more efficient parallelization when sampling in GRAPES.

---

### Official Review · Reviewer_4YMr · 2023-11-01

**Soundness:** 4 excellent
**Presentation:** 4 excellent
**Contribution:** 4 excellent
**Rating:** 8
**Confidence:** 4

**Summary:**

This paper introduces a novel mechanism for sampling subgraphs from a large graph during GNN training, with the goal of increasing scalability.

This sampling mechanism samples a fixed number k of nodes to add to the subgraph at each layer following a reinforcement learning policy parametrized by a GFlowNet and optimized using a trajectory balance loss. In addition to this framework, a key contribution of the paper is employing the loss of the GNN in the downstream task as the reward. Therefore, the model learns to sample nodes adaptively, in a manner that improves performance in the downstream task.

The efficacy of the architecture is verified via extensive numerical experiments on small, moderate-size, and large graph node classification tasks.

**Strengths:**

- The motivating ideas for the proposed sampling framework are very novel. They are a great example of integrating different concepts/techniques from modern deep learning to solve a relevant problem --- scalability of GNNs.
- Although this is primarily an algorithm-based/application paper, the model, the algorithm, and the training mechanism are theoretically grounded, and the authors did a very good job at motivating and explaining the reasons behind their design choices.
- The numerical results are extensive and convincing. I appreciate the inclusion of hypothesis tests for the rank of their method with respect to the baselines; the memory plots comparing GRAPES with GAS; the transferability plots of performance versus subgraph size; and the entropy plots. In particular, the transferability plots (Fig. 3) are very convincing in showing the superiority of GRAPES, as its performance is much more robust to reducing K. Further, the entropy plots are in direct agreement with the authors's claim that GRAPES is consistent in identifying important nodes.

**Weaknesses:**

- Some related work is missing, and perhaps also a comparison with other graph sampling baselines from the graph signal processing literature. Check, e.g., "Efficient Sampling Set Selection for Bandlimited Graph Signals Using Graph Spectral Proxies", by Anis and others, and papers therein (specifically, the works of Kovacevic and Moura; Chamon and Ribeiro; Segarra, Marques and Ribeiro; etc.). These papers are part of a subfield of graph signal processing---graph signal sampling---which studies how to sample graphs so as to maximize the preservation of their spectra. Since graph spectral information is typically very correlated with performance in graph machine learning tasks, I believe these are important references/comparisons to include.
- The explanation of why the method is trained off-policy is not very clear for readers not familiar with reinforcement learning. There is a result which is only mentioned in passing---"Importantly, GFlowNets [...] can learn from off-policy distributions without adjusting the objective"---which is important in justifying the choice of off-policy training, and hence should be described in further detail (perhaps a short subsection) in the camera-ready. It would also be interesting to see empirical comparisons between training off-policy and using gradient estimation methods.
- The numerical experiments only consider node classification tasks.
- Other relevant line of related work is that on the "transferability properties of GNNs". See e.g. the work of Ruiz et al.

**Questions:**

- Have you analyzed the specific subgraphs that are sampled by GRAPES in different tasks? What are their characteristics (are they connected? do the sampled nodes have high centrality? etc.). GRAPES sounds like a nice tool for understanding which characteristics of a graph are most important in a given task.

---

> ### Author Response · Authors · 2023-11-17
> **Response to the Comments from Reviewer 4YMr**
>
> Thank you for your feedback. Here are our answers to your comments and questions.
>
> 1. Graph signal processing literature: Thank you for the pointers; indeed, relevant works. The mentioned paper looks into sampling nodes in a graph for a unique and stable graph signal reconstruction. This work has been used in [1] as a node sampling technique for graph neural networks. We included these works in the related work of our revised paper. Also, the study on “transferability properties of GNNs” [2] uses the weights of a GNN trained on a mid-sized graph and transfers them to larger graphs given the graphon similarity between the graphs. Although this work doesn’t focus on sampling, it’s an interesting approach to scaling GNNs to larger graphs. We included this paper in the related work of our revised paper.
> 2. On expanding on the off-policy setup: This is a fair point. We changed the description of why Trajectory Balance has favorable properties according to the literature. The sentence we changed is now: “[3] showed that the trajectory balance loss is stable when learning from off-policy distributions without adjusting the objective with importance weights”. We also clarified Section 4.3 and added a larger Appendix section (Appendix A) going into more detail for our off-policy setup, discussing the challenges with computing importance weights for RL being intractable.
> 3. Only node classification: Our method is general enough to support many tasks. We focused on node classification, which already required running a significant number of experiments. However, exploring GRAPES in other tasks is a good direction for future work.
> 4. Analysis of the sampled nodes: This is a very good question. We plan to focus on this in future work. We believe that evaluating GRAPES on knowledge graphs, which have different relation and node types and multimodal information, can give us more insight about the type of information that is influential in the graph for a GNN.
>
> ### References
> [1] Geng H, Chen C, He Y, Zeng G, Han Z, Chai H, Yan J. Pyramid Graph Neural Network: A Graph Sampling and Filtering Approach for Multi-scale Disentangled Representations. InProceedings of the 29th ACM SIGKDD Conference on Knowledge Discovery and Data Mining 2023 Aug 6 (pp. 518-530).
>
> [2] Ruiz, L., Chamon, L.F. and Ribeiro, A., 2023. Transferability properties of graph neural networks. IEEE Transactions on Signal Processing.
>
> [3] Malkin, N., Lahlou, S., Deleu, T., Ji, X., Hu, E. J., Everett, K. E., ... & Bengio, Y. (2022, September). GFlowNets and variational inference. In The Eleventh International Conference on Learning Representations.

---

### Official Review · Reviewer_uctj · 2023-11-01

**Soundness:** 2 fair
**Presentation:** 1 poor
**Contribution:** 2 fair
**Rating:** 3
**Confidence:** 2

**Summary:**

The work proposes a new classification model for large-scale graph data. It combines a learnable node-selection component and a graph neural network to construct a unified classification model. The node-selection component is devised with a GFlowNet, which is further parameterized by a graph neural network. Essentially it is an RL learning component that helps minimize the training loss. With this construction, it can greatly reduce the number of nodes in the classification component but without much performance drop.

**Strengths:**

Compared with previous models such as AS-GCN, the sampling method of this model is more "adaptive". Although the model is much more complex than previous models, it does show some performance improvement.

**Weaknesses:**

1. The description of the algorithm is problematic. As far as I know, GFlowNet is a method proposed to sample for an energy-based model: it addresses a distribution approximation problem. It is a special case of an RL algorithm. In my view, the paper is a pure RL problem especially since the reward function is clearly defined. I think an RL formulation is straightforward from that. The formulation with GFlowNet is very misleading -- I spent hours before realizing that this is not a distribution approximation problem.  Actually, the reward scaling in 4.2 could be avoided within an RL formulation.

2. The method is much more complex than previous methods because it has this extra learnable component. I don't know whether it is easy for others to apply such a model to a different application. To me, the simplicity of model tuning is more appealing than minor performance improvement: one may not see the improvement if the model cannot be well-tuned.

3. The performance values of baseline methods reported in Table 1 are much lower than those reported in their original papers. I don't know how much I can trust the comparison. For example, Graph-SAINT has f1 scores, 0.511±0.001, 0.966±0.001,  and 0.653±0.003 on the Flickr, Reddit, and Yelp datasets. These numbers are much higher than the reported numbers in the submission.

**Questions:**

On the ogbn-products dataset, can you tune the number of samples (n) so that AS-GCN can also run on this dataset?

Can you put data statistics in the experiment section? These numbers are important to the understanding of experiment results.

---

> ### Author Response · Authors · 2023-11-17
> **Response to the Comments from Reviewer uctj**
>
> Thank you for your feedback. Here are our answers to your comments and questions.
>
> 1. Why we use GFlowNets instead of RL: This is a great question with a nuanced answer. The main difference is in the objective the two approaches aim to optimize. Roughly, RL is about finding a policy that minimizes the loss and converges to a deterministic solution. Instead, GFlowNet aims to sample in proportion to low loss and usually converges to a stochastic solution. The reviewer is correct that GFlowNets are a special case of RL in that recent theoretical results show GFlowNets optimizes a maximum-entropy RL objective (see e.g. [1], Section 7.2).
>
>    The second reason why GFlowNets are useful in our setting is more technical: Its off-policy properties. RL requires a policy distribution. In setup, we need a distribution for sampling exactly $k$ out of $n$ nodes. This is a tricky distribution that scales polynomially in computation in both $k$ and $n$ [1] (we tried to use the codebase of [2], but this failed already in medium-sized graphs). Instead, we used a distribution that is a product of independent Bernoulli’s and used the Gumbel-max trick to sample exactly $k$ positive values from this distribution. This is off-policy: We sample from a different policy than the distribution we optimize. In GFlowNet loss functions, we can use any off-policy distribution without changing the loss, while in RL loss functions, off-policy sampling requires importance weighting, which would again require computing the distribution presented in [2].
>
>    We modified the paper to highlight why we chose GFlowNets instead of (regular) RL. (Section 2.2), and added extra explanations on the off-policy sampling in Section 4.3 and Appendix A.
>
> 2. Using a more complex method: We agree with the reviewer that our model is more complex than the heuristics-based methods, i.e., FastGCN, LADIES, and GraphSAINT. However, the accuracy improvement of GRAPES compared to these baselines is not minor. We argue that to have an adaptive method, one needs a learnable component in sampling, which necessarily adds complexity. The same happens with AS-GCN, which needs to train an attention layer for sampling and uses significantly more memory than GRAPES. About the reviewer's concern about the other applications of GRAPES, the design of our model is not dependent on any specific architectures or loss functions; therefore, it can be easily applied to other tasks like link prediction.
>
> 3. The baselines’ performance was lower than reported in the original papers: Each baseline used different GNN architectures, sampling rates, batch sizes, etc., which makes it difficult to judge if the reason for their good performance is the sampling method or the mentioned different factors. Therefore, for a fair comparison, we kept all the other factors the same for all the methods. For GraphSAINT, the major changes were the GNN architecture (from two higher order aggregator layers [3, 4] to two GCN layers), the number of nodes per batch (from thousands to $BatchSize+2*SampleSize=256+256+256$ nodes), and the depth of sampling (for some datasets from 4 to 2 layers of depth). Using this uniform setup, where only the sampling method varies, we can clearly show that GRAPES as a sampling method outperforms the baselines.
>
> 4. Dataset statistics: Because of the lack of space in the main body of the paper and the nine-page limit, we put this in the appendix.
>
>
> ### References
> [1] Bengio, Y., Lahlou, S., Deleu, T., Hu, E. J., Tiwari, M., & Bengio, E. (2023). Gflownet foundations. Journal of Machine Learning Research, 24(210), 1-55.
>
> [2] Ahmed, K., Zeng, Z., Niepert, M., & Van den Broeck, G. (2022, September). SIMPLE: A Gradient Estimator for k-Subset Sampling. In The Eleventh International Conference on Learning Representations.
>
> [3] Abu-El-Haija, S., Perozzi, B., Kapoor, A., Alipourfard, N., Lerman, K., Harutyunyan, H., Ver Steeg, G. and Galstyan, A., 2019, May. Mixhop: Higher-order graph convolutional architectures via sparsified neighborhood mixing. In international conference on machine learning (pp. 21-29). PMLR.
>
> [4] Hamilton, W., Ying, Z. and Leskovec, J., 2017. Inductive representation learning on large graphs. Advances in neural information processing systems, 30.

---

> > ### Comment · Reviewer_uctj · 2023-11-22
> > **Thank you for your clarification**
> >
> > For the first question: if you only need an off-policy learning method, you don't need to take the entire language of GFlowNet because RL has plenty of off-policy learning methods. I feel that an accurate description of the problem and the solution is important for readers' understanding and future work that will follow yours.
> >
> > For the third question: given the complexity of current neural networks, your implemented version may not be as good as the original version. If there is released code for baseline methods, a better practice is to use their released code.

---

> > > ### Author Response · Authors · 2023-11-23
> > > **Second response**
> > >
> > > We thank you for your response.
> > >
> > > To answer your first point, we emphasize that we do not simply require any off-policy method. As noted in our first answer, using RL off-policy methods requires importance weighting, which is intractable for our problem. We rely on the properties of GFlowNets to allow us to train efficiently off-policy. Secondly, as mentioned, using RL would only minimize loss, while we’re interested in sampling a variety of solutions. This is a different task with a different set of solutions.
> > >
> > > In response to the second point, we used the released code by the authors of the baselines, and made every effort to ensure a fair comparison between methods. For example, we removed the attention mechanism in the classification step of AS-GCN, because it highly affects the F1-score, making it difficult to judge the effect of sampling on the performance. Also, in GraphSAINT, we lowered the high sample size that significantly increases the F1-score. These features must be the same among all the methods to eliminate their effect and to solely determine the impact of sampling. The key issue here was not that baseline code was not available, but that we require a uniform setup to fairly compare these sampling methods.

---

> > > > ### Comment · Reviewer_uctj · 2023-12-02
> > > > **Off-policy learning does not require GFlowNet**
> > > >
> > > > "using RL off-policy methods requires importance weighting,"
> > > >
> > > > This is incorrect: if you don't require the gradient to be unbiased, you don't need importance sampling. In fact, GFlowNet uses biased estimations of gradients and does not require importance weights. It doesn't mean that the GFlowNet framework is necessary for off-policy learning. I feel that this saying is misleading.

---

### Author Response · Authors · 2023-11-17
**General Response to the Reviewers**

We would like to thank the reviewers for their valuable comments. The reviewers highlight the novelty of integrating the GFlowNet framework in GNNs (J9DQ, 4YMr). They appreciate that the paper is to the point, easy to follow (iUex, J9DQ), and mathematically grounded (4YMr). We value the reviewers' attention to our experiments supporting our claim about GRAPES robustness to low sample size and its varying preference over nodes. Please find our answers to the questions below each review.

---

### Meta-Review · Area_Chair_nrRv · 2023-12-15

**Metareview:**

The paper presents a novel subgraph sampling mechanism for enhancing scalability in Graph Neural Network (GNN) training. The approach involves sampling a fixed number of nodes at each layer, guided by a reinforcement learning policy parameterized by GFlowNet and optimized with trajectory balance loss. Notably, the GNN loss in the downstream task serves as the reward, enabling adaptive node sampling that enhances performance. Experiments across small, moderate-size, and large graph node classification tasks validate the efficacy of the proposed architecture.

The reviewers agreed that the core idea is novel. However, several reviewers had concerns about the experiments, specifically the comparison to prior work as well as about the use of GFlowNet. Some reviewers question the reliability of the performance values reported in Table 1, noting that baseline method results are considerably lower than those in their original papers. Furthermore, modifications to the baseline architecture raise concerns about the validity of the comparison to previous work. These reviewers recommend running the original code of baseline methods for a more meaningful comparison.

There was a discussion between the AC and the reviewers after the rebuttal phase. While one reviewer was strongly in favor, the other reviewers were more negative and actively argued that the paper has significant weaknesses. This was also true for the reviewer who scored the paper with a 6. They were also more critical after the rebuttal. Hence, we believe that the paper is not ready for acceptance. Possible improvements are more rigorous treatment of baselines in the experiments and stronger arguments for using GFlowNet.

**Justification For Why Not Higher Score:**

Several reviewers argued actively against accepting the paper due to substandard experiments and missing justifications for the method.

**Justification For Why Not Lower Score:**

There is no lower score

---

### Decision · Program_Chairs · 2024-01-16

Reject